# A Predator on the Doorstep: Kill Site Selection by a Lone Wolf in a Peri-Urban Park in a Mediterranean Area

**DOI:** 10.3390/ani13030480

**Published:** 2023-01-30

**Authors:** Marco Del Frate, Paolo Bongi, Luigi Tanzillo, Claudia Russo, Omar Benini, Sara Sieni, Massimo Scandura, Marco Apollonio

**Affiliations:** 1Department of Veterinary Medicine, University of Sassari, 07100 Sassari, Italy; 2Department of Veterinary Science, University of Pisa, 56100 Pisa, Italy; 3Department of Agricultural Management, Food and Forestry System, University of Florence, 50100 Florence, Italy

**Keywords:** *Canis lupus*, predatory strategy, predation, body condition, landscape elements

## Abstract

**Simple Summary:**

Wolves are known for their cooperative hunting behaviour, which has been thoroughly studied. On the contrary, little is known about lone wolves’ hunting strategies. Between 2017 and 2019, we monitored a lone male wolf living in a Mediterranean coastal area. The wolf settled in a protected estate where both wild and domestic ungulates were available as potential prey. His predatory behaviour was recorded through a combination of camera trapping and active search for kill sites and prey carcasses. Features of kill sites were modelled to test for selection by the wolf. The main prey resulted to be fallow deer. The wolf avoided dense habitats to hunt and usually attacked his prey in open areas where the presence of fences could support him in constraining and successfully killing deer. The prey body condition, estimated as the percentage of fat content in the bone marrow of hind legs, was above the average of the population, suggesting a high efficacy for the lone wolf hunting strategy but also the adoption of a high-risk feeding strategy by deer.

**Abstract:**

The aim of the study was to assess which kill site characteristics were selected by a lone wolf living in a protected Mediterranean coastal area near the city of Pisa, Italy, where both wild and domestic ungulates were available as potential prey. Between 2017 and 2019, we monitored the wolf’s predatory behaviour through a combination of camera trapping and active search for kill sites and prey carcasses. The main prey found was the fallow deer (n = 82); only two wild boars and no domestic ungulates were found preyed upon. The features and habitat of kill sites were modelled to test for selection by the wolf. The habitat type of kill site was composed of meadows and pastures (89.3%), woods (7.3%), degraded coastal areas (1.9%), roads and rivers (1.1%), and marshes (0.5%). We calculated their distance from landscape features and ran a binomial generalised linear model to test the influence of such landscape variables. The distance of kill sites from landscape elements was significantly different from random control sites, and a positive selection for fences was found. In fact, the wolf pushed fallow deer towards a fence to constrain them and prevent them from escaping. We also analysed the body condition of predated fallow deer as a percentage of fat content in the bone marrow of the hind legs. Our results revealed the selection of the lone wolf for deer in good body condition. This is a possible outcome of the habitat selection shown by fallow deer in the study area, where fenced open pastures are the richest in trophic resources; therefore, our findings suggest a high efficacy for the lone wolf hunting strategy, but also the adoption of a high risk feeding strategy by deer. This study suggests that a lone predator can take advantage of human infrastructures to maximise its predatory effectiveness.

## 1. Introduction

In predator–prey systems, kill sites are often not randomly distributed [1,2,3,4], as environmental factors, both man-made and natural, can influence prey detection, access, and finally, the attack success [2,4,5,6].

Large carnivores’ predatory behaviour has been well investigated [7,8,9,10,11,12], and for the wolf (*Canis lupus*), group hunting is shown to enhance the efficacy of searching, approaching, and attacking prey [12].

Accordingly, the selection of kill sites represents a crucial aspect of hunting behaviour [13]. MacPhee and colleagues [4] pointed out how wolves preferred to hunt their prey at sites where the probability of a prey encounter was higher. On the other hand, high prey densities could have a detrimental impact on the wolf’s selection of kill sites because they would make deer vigilance higher and reduce the effectiveness of the wolf attack [14].

While the hunting behaviour of wolf packs is well studied [7,15,16,17], the same is not true for small groups (pairs or three individuals), and even less data are available on solitary wolves [18,19,20]. Solitary wolves frequent large areas, very similar to those used by large packs (more than three individuals [20]). Being lone does not seem to affect the ability to kill prey, as reported by Thurber and Peterson [20] “*...solitary wolves and pairs readily killed adult moose, in contrast to a common belief that larger packs benefit from cooperative hunting*”. However, this statement refers to moose (*Alces alces*) predation and does not specifically consider which predatory strategy was adopted and at which site the killing occurred.

Moreover, habitat characteristics could influence prey choice; for instance, in open habitats or in even-aged high forests with limited undergrowth, cervids constituted the principal component of the wolf diet [21,22,23], while wild boar (*Sus scrofa*) was preferably hunted in dense forests [24].

Tall vegetation and scrubland were the most important habitat types that influenced the risk of livestock predation by wolves in Mongolia [25]. In general, livestock seemed more vulnerable to wolf predation in areas with dense vegetation [26,27,28]. From this perspective, predation activity could be considered as a hierarchical process [2,4,12,29] where wolves select hunting areas, and within them, killing sites are chosen according to prey density and landscape elements [4], or may be influenced by habitat characteristics regardless of prey density [14].

However, wolves not only select killing sites but also select the individuals to prey upon within the population. In general, wolves kill mostly young (<12 months) ungulates, while among adults, their predatory activity is often targeted at debilitated or sick individuals [30,31,32].

However, the wolf is an opportunistic predator and can also use anthropogenic resources [33]. In Europe, wolf presence has been increasing in the last decades [34], and consequently, their interactions with humans both in rural [26,35,36] and in urban areas [33,37] have become more frequent.

In urban and peri-urban areas the main resources used by wolves were reported to be garbage or remains of slaughtered livestock [37,38,39,40]; however, they could attack and kill livestock [41,42] and pets [26,43,44], especially dogs, by using them as a food resource [45,46]. Furthermore, in Europe during the past decades, ungulates have moved closer to urban areas due to their significant increase [47] and the spread of urban settlements [48]. As a consequence, wolves have moved to peri-urban areas as well [49], following their main prey that in the European continent are currently represented by wild ungulates [24,50,51,52].

The main goal of this study was to investigate the kill site characteristics and prey body condition in a peri-urban protected area occupied by a lone wolf and hosting wild and domestic ungulate populations.

We aimed to verify the following:(i)if the wolf selected or avoided specific habitats to attack its prey;(ii)which landscape elements can influence the choice of the predation site;(iii)which was the body condition of the prey, by using leg bone marrow fat analysis as an indicator of ungulate nutritional status (as, for example, in red deer [53] and white- tailed deer [54]).

## 2. Materials and Methods

### 2.1. Study Area

We conducted the study in the Migliarino–San Rossore–Massaciuccoli Regional Park, specifically in the San Rossore Estate (43°41′ N, 10°19′ E; Figure 1), a protected area of 47.0 km^2^, 5 km from the city centre of Pisa (about 90,000 inhabitants), Italy. The estate is delimited to the north by the Serchio River, to the south by the Arno River, to the west by the sea, and to the east by a fenced line that borders the city of Pisa. Two additional rivers, namely Fiume Morto Nuovo and Fiume Morto Vecchio, cross the area (Figure 1).

The study area is characterised by a sub-Mediterranean climate that is covered by mixed deciduous forest (*Quercus robur*, *Q. ilex*, *Fraxinus* spp., *Populus alba*, *Carpinus betulus*) and pine forest (*Pinus pinaster* and *P. pinea*), alternating with marshes (*Carex* spp., *Phragmites australis*, *Juncus* spp.) and meadows (see [50] for detailed description of habitat composition in the study site). The wooded areas represent 67.4% of the whole estate, while open areas amount to 32.6%.

In the estate, there is still limited agriculture and livestock breeding in fenced areas totalling about 7 km^2^. The inner areas of the estate (close to the sea) are essentially off-limits to visitors during daylight hours. Two ungulate species live in San Rossore: fallow deer (*Dama dama*) and wild boar, and the gamekeepers manage these ungulates populations.

According to the vantage point counts carried out throughout the estate and the consequent management plans, between 1984 to 2004 [55], spring fallow deer density averaged 29.0 deer/km^2^ (SD = 5.5 deer/km^2^). In recent years, this value has increased, reaching an estimated spring density of 49 deer/km^2^ in 2019 (Management Plans of the Migliarino–San Rossore–Massaciuccoli Regional Park). According to management plans, only park guards are authorised to conduct management tasks on ungulates, including fallow deer culling.

The wild boar population experienced less variation, as the spring density estimate ranged from 2.5 to 8 heads/km^2^ in the same time span. Horses (1.4 heads/km^2^) and cows (3.3 heads/km^2^) are the only domestic ungulate livestock present in the estate, at the moment of the study.

Since spring 2016, a lone wolf has been reported in the San Rossore Estate, and until February 2019, no other wolves were observed in the area. The wolf pack closest to the study area was located about 18 km to the south [56,57,58].

On the estate, there are other carnivores such as red fox (*Vulpes vulpes*), badger (*Meles meles*), weasel (*Mustela nivalis*), pine marten (*Martes martes*), and stone marten (*Martes foina*).

### 2.2. Field Methods

From March 2017 to February 2019, we monitored the wolf’s presence through transect and camera trapping surveys. We selected transects at random, encompassing the majority of the study area; some transects coincided with nearby roadways, while others crossed the vegetation (bushes or trees). Transects, totalling 104.8 km (8.7 km on average for each transect), were walked twice a month throughout the duration of the study activity in search of wild ungulate carcasses to track wolf predations.

We randomly distributed transects in the study area, and 24.6 km (corresponding to 23.5% of the total distance) were in open areas, while 80.2 km (76.5%) were in wooded areas.

We employed twenty camera traps (model Scout Guard BG960 with motion sensor, 24 MP images, 1080 p video, 120° wide-angle lenses, 30.5 m lighting ranges, and trigger times of ≤0.7 s) for the camera trapping operations that were always on (24/h) and in video mode. Camera trapping monitoring started on 4 March 2017, and concluded on 25 February 2019. Camera traps were active in the field for 723 days, for a total of 14,106 camera-trap days, with an average of 705.3 days per camera (min. 654 days; max 723 days).

The study area was divided into 20 regular polygons of 2.35 km^2^ each, and in each polygon, the camera site was selected with an opportunistic approach.

Predation was considered the cause of death when blood, subcutaneous haemorrhaging at wound sites or signs of struggle were found at the site. We considered only recent predation events, evaluated on the presence of fresh blood and carcasses not yet consumed.

We recorded species, sex and age classes of the prey based on analysis of teeth eruption/consumption. In addition, for the most frequently preyed species, i.e., fallow deer, a hind foot was removed from each carcass, and stored in polythene bags at –5 °C, to assess the body condition of the prey through the analysis of the long bone marrow fat [59]. As a reference for the body condition, during the whole study period, we collected with the same method one hind leg from deer of the same sex and age class culled by park guards in the San Rossore Estate.

### 2.3. Geographic Information

We registered the location of carcasses on a GPS information system using Garmin 60 CSx, taking coordinates in the UTM system Zone 32 North. We also generated 1000 random points as “control sites” using QGis software to compare the characteristics of the kill sites with a set of random sites. Since transects covered the whole study area, we selected random points in the estate in order to achieve a consistent and representative survey; 73.7% of the random points were in wooded areas, while the remaining 26.3% were in open areas. Vegetation and other environmental attributes were analysed for each site (predation and control) from geographic information system (GIS) land cover map layers (50 × 50 m resolution) (Tuscan Regional Government—https://www502.regione.toscana.it/geoscopio/cartoteca.html, accessed on 7 July 2022). We measured habitat variables in a 50 m radius plot centred on the predation site and at control sites. We chose a radius of 50 m because, based on the analysis of the camera trap videos, no prey was moved to a further distance from the kill site.

Using Garmin 60X GPS, we georeferenced every fence inside the study area and drew it digitally. For each point, we considered the percentage of the following habitat types: meadow and pasture, deciduous wood, coniferous wood, marsh, degraded coastal area, river, human settlement, and road. Furthermore, we took into consideration the distance (in meters) from fences, buildings (mostly uninhabited houses or ruins), roads, rivers, and the edge of the forest. They represent the most significant elements to prevent or constrain the prey’s flight.

### 2.4. Wolf Identification

With the aim of identifying a single wolf in the area, we searched for fresh scats on transects; once found, a scat was stored in a plastic bag and frozen at –20 °C. Then, we analysed it to determine sex, source population, and individual profile, using a set of markers including the amelogenin gene, the mitochondrial control region (mt-CR), two Y-linked and 11 autosomal microsatellites, following the methodology described by Canu et al. [60]. We classified the 350-bp mt-CR sequence obtained using the BLAST search engine (https://blast.ncbi.nlm.nih.gov accessed on 7 July 2022) in order to ascertain whether it matched with the diagnostic Italian wolf haplotype W14 [61]. Similarly, we checked if the Y-linked microsatellite alleles matched those previously observed in the Italian wolf population [62]. We compared the 11 microsatellites profile with a database of >400 wolf genotypes from the same region (unpublished data) to check if it had previously been sampled in another area of Tuscany.

Phenotypical identification was based on the comparison of morphological traits of the wolves recorded in the videos with the aim of detecting individual-specific characteristics (Appendix A).

### 2.5. Laboratory Activity

To assess the body condition of predated individuals, we collected the metatarsus from a hind leg of the carcass and stored it frozen. To proceed with bone marrow extraction, a 40 mm section of marrow was removed from mid-way along each bone, weighed to 0.0001 g, and oven-dried at 80 °C to constant weight (±0.02 gm). We weighed the marrow again following the oven-drying procedure, and we expressed the fat content as a percentage of the initial weight [63].

### 2.6. Data Analysis

In order to investigate which environmental variables influenced wolf selection for predation sites, we performed binomial logistic regressions on what was “used” by the wolf. The availability (percentage coverage within a 25 m buffer) of six different habitat types (coniferous wood, deciduous wood, marshes, meadows, degraded areas, roads, and river) were included as continuous predictors to evaluate the effect of habitat composition in the surroundings of the predation site selected by the wolf. Other continuous predictors were: distance from the nearest wood, from the nearest river, from the nearest fence, from the nearest building, and from the nearest road (m). We screened all predictors for collinearity (Pearson coefficient |rp| < 0.7) and multicollinearity (variance inflation factor, VIF < 3, [64]). The distance from the nearest fence was in a non-negligible collinear relationship with the other three predictors (meadow availability, distance from the nearest building, and the distance from the nearest river, rp= −0.7, 0.7, and 0.7, respectively). Thus, we performed a random forest calculation (randomForest R package) to rank predictors on the basis of their potential to explain the variation in the amount of use. Since the distance from the nearest fence had the highest rank across the collinear predictors, we dropped the meadow availability, the distance from the nearest building, and the distance from the nearest river from subsequent analyses.

We used the “glm” R function to run a binomial GLM (generalised linear model), with “used” as the response variable. We subsequently ran a set of models with all possible combinations of the predictor variables included in the full model by means of the “Dredge” function (MuMln R package). We selected the best model following the minimum AIC criterion [65]. In the case of models with ΔAIC < 2, we selected the most parsimonious in terms of the number of predictor variables included [65].

Furthermore, we used distances from landscape elements as independent variables in a binary logistic regression model, where 1 indicated a wolf predation site and 0 represented a random point. This model was used to determine if the wolf was selecting specific site attributes at predation sites versus randomly generated sites (control sites). We computed a Hosmer–Lemeshow statistic to test the model goodness-of-fit (GOF), rejecting the null hypothesis of selection proportional to habitat availability in the presence of low *p*-values (< 0.05).

We tested the distances from different landscape elements for differences among seasons, using multivariate analysis of variance (MANOVA).

To test for a selection of prey based on their body condition, we used the percentage of the dry weight of bone marrow fat (BMF) contained in the metatarsi (%BMF) as the dependent variable in an univariate analysis, with sex and age class as independent variables. Specifically, to test the hypothesis of wolf selection of prey in poor physical conditions, we compared the %BMF of predated fallow deer with the %BMF of fallow deer randomly culled within the annual management plan (control). The Student’s t-test was performed to test their difference for significance (*p* < 0.05).

We performed all analyses using R software [66].

## 3. Results

### 3.1. Wolf Identification

A single wolf was detected in the area during the study period. The analysis of a single fresh scat confirmed that it was a male from the Italian population (it carried the W14 mt-CR haplotype, private to the Italian population [61], and the H2 Y-chromosome haplotype, the second most common in the Italian peninsula [62]) and had not been sampled before in other areas of Tuscany. During the whole study period, each camera recorded at least one video of the lone wolf, for a total of 439 videos of this wolf, and we recognised him by morphological traits in 384 videos (87.5%). As a matter of fact, this individual had an atypical morphology that enabled its recognition: it had a short tail with an abnormal curve at one-fourth of its length (possibly due to trauma or malformation) and whitish front feet (Appendix A). Most individuals recorded had at least one of these characters visible in the video, and in all cases, it matched the above-described pattern. The remaining videos were not assigned, because none of these morphological clues were detectable (i.e., the tail and feet were not clearly visible in the video). Our data allowed us to confirm the presence of only one wolf; if another wolf had frequented the estate and actively participated in the fallow deer predations, it would have been observed and identified at least once through camera trap monitoring.

### 3.2. Kill Sites Characterisation

We found and analysed 84 kills and relative predation sites detected during the study period. Almost all predations were on fallow deer (82 individuals) belonging to different age and sex classes, and only two were on juvenile wild boars. No predation on domestic ungulates was reported in the estate in the same period.

We evaluated the habitat composition of the kill site in a circular buffer area with a 50 m radius (Table 1). The large majority of the kills (95.1%) were found in open areas, most of them in proximity to fences (Appendix A).

All models performed with the resource selection function, including habitat type and distances from landscape elements (parameter estimates reported in Table 2), showed a negative selection for wooded areas. Since the variables “meadows and pasture” and “distance from the fence” were strongly correlated with each other, they were not included in the best model selected on the basis of AICc values.

The distance from landscape elements was significantly different between predation sites and random sites (RSF model: Z_Wood_ = −3.794, *p*_Wood_ = <0.001; Z_Roads_ = −4.196, *p*_Roads_ = <0.001; Z_Building_ = −2.758, *p*_Human_settlements_ = 0.103; Z_Fences_ = −3.968, *p*_Fences_ = <0.001). The wolf selected areas near fences to kill fallow deer (Table 3), and fences were the closest elements in all seasons (Table 4).

The logistic regression model underlined the differences in the distances from the landscape elements considered (Table 5), and it showed a positive selection for fences, whereas roads were avoided.

The minimum distance from fences did not show significant differences among seasons (MANOVA: F = 1.597, *p* = 0.125), while the distances from the other elements were different depending on the season considered (MANOVA: F_Building_ = 2.150, *p*_Building_ = 0.031; F_River_ = 3.110, *p*_River_ = 0.002; F_Road_ = 3.563, *p*_Road_ = 0.001; F_Wood_ = 2.602, *p*_Wood_ = 0.009).

### 3.3. Prey Characteristics

We evaluated the body condition of dead fallow deer as %BMF and the fat content in the metatarsus was higher in fallow deer preyed upon by the wolf than in fallow deer culled by gamekeepers (Student’s *t*-test: t = 2.159, *p* = 0.033, Figure 2); thus it suggests that the predated deer were in good physical condition, at least as the control ones.

Moreover, the %BMF differed among age classes of deer predated by the wolf (ANOVA: F = 9.114, *p* < 0.001; Table 6). In particular, fawns showed the worst body condition (on average 36.3 %BMF) with a statistically significant difference between sexes (males: 22.0%; females: 50.7%; Student’s *t*-test: t = −2.504, *p* = 0.025). Conversely, individuals aged 1–2 years exhibited no sexual difference in %BMF (75.8% for males and 68.1% for females; t = 0.531, *p* = 0.276), same as individuals between 2 and 4 years of age (73.5% for males and 66.5% for females; t = 0.185, *p* = 0.478).

## 4. Discussion

The present study highlights how a lone wolf settled in a suburban protected area has found extremely favourable trophic conditions, where fallow deer have become its main prey. We monitored the lone wolf for more than two years, gathering 439 wolf videos in which we were able to identify the wolf based on specific anatomical traits [60]. Additionally, genetic analysis confirmed it was a male from the Italian wolf population, exhibiting a Y-chromosome haplotype very common in the Italian peninsula, so the most likely hypothesis was that it represented a natural immigrant (but we could not assess the exact source area because of limited reference genotypes and weak genetic structure).

It is unusual that a wolf ends up living alone and stays in one area for so long. Similar behaviour has been observed in Canada, where an island-bound wolf adapted its diet and reduced its home range to fit the island confines [67]. Even if the lone wolf does not meet the social demands of the species, the high density of deer and the lack of competition in San Rossore Estate likely promoted the wolf’s persistence in the area.

The wolf seems to have developed a predation strategy based on the aid offered by fences that allowed him to catch his prey more easily.

We think that our dataset could be a subsample of all prey actually killed by the wolf during the study period, but the uniform distribution of the effort in the different areas of the estate allows us to state that the killing sites were selected on the basis of specific characteristics.

### 4.1. Factors Affecting Kill Sites

We found a high percentage of kill sites in open habitats (meadows and pasture), in agreement with other studies that considered deer species as prey [13,23], but not all open areas investigated contained fallow deer kills. Our research revealed the importance of man-made elements, namely fences, in choosing kill sites. As a matter of fact (see Appendix A), the wolf pushed the fallow deer towards a fence to constrain it and prevent it from escaping, adopting a hunting strategy also recorded by Bojarska and colleagues in Western Poland [68]. During our field surveys, we often found bulges in the nets right at predation sites. On the other hand, the fences in our study area are located in correspondence with the grazing areas, i.e., open habitats. Even if open habitats allow higher visibility and allow prey to spot the predator’s arrival [69], the presence of fences can, in turn, prevent their escape and increase the success of the wolf’s attack.

For these reasons, kill sites were concentrated in two sectors of the study area, i.e., in the north and in the south, the only areas where a combination of open areas and fences was present (see Appendix A). These two areas became the most favourable places in the estate where a single wolf (30–35 kg) could be able to constrain and kill prey as big as a 100 kg buck. In other zones of the study area, an unconstrained fallow deer had many flight routes and a high chance of escaping the attack. In fact, very few kills were found in the rest of the study area.

The walked transects were representative of the study area, as they covered the full estate and crossed (as mentioned above) open and closed habitats in similar proportions as they occurred in the estate. Therefore, in the absence of selection by the wolf, the expected probability of finding a kill along transects should have been much higher in forest than in open areas, being forests three times more abundant than open areas. Moreover, among open areas, those without a fence were more frequent than fenced ones, so more kills were expected in the former.

On the contrary, the observed distribution of kills was the opposite, as fenced open areas had the majority of predations, followed by unfenced open areas and then by forest, where no kills were found. It should be underlined that forests in the study area had no or very limited understory as a consequence of fallow deer overabundance that drove to overgrazing, therefore, visibility was definitely high, comparable to grass pastures present in open areas.

The orography of our study area confirmed that flat areas represent optimal hunting grounds for wolves [3]. In contrast with Gula [70], our results show that creeks and/or river beds were not preferred killing sites, as they do not necessarily represent an obstacle to escape; indeed, deer are used to cross creeks and rivers in the area during their daily movements [71].

As, during our study, the wolf lived alone in the area, it hunted without any conspecific helpers and therefore developed a specific hunting technique implying the use of fences to block his prey. In so doing, the wolf might have replaced the collaboration of pack mates [12].

Intense human use of the area did not seem to limit the wolf’s activities: even though the estate is located near a city (Pisa) and many people visited the estate every day, the wolf attacked his prey in the sectors with high human presence [71]. However, its predatory activity took place mostly at night and dawn, making it possible to avoid human presence even in such a crowded area.

### 4.2. Prey Characteristics

The wolf preyed only on wild ungulates and, in particular, fallow deer, which is the species with the highest density in the estate, confirming an opportunistic behaviour, i.e., use of the most available prey [21,24,50,52,72].

Body condition of preyed fallow deer, as recorded from the marrow fat content in leg bones, was good, as already observed in other studies (e.g., white-tailed deer *Odocoileus virginianus*: [73]; moose: [74]; red deer *Cervus elaphus*: [21]. As matter of fact, the wolf’s prey displayed a percentage of marrow fat that indicated it was in good physical condition, comparable to the other fallow deer culled in the estate. This was in contrast with other studies where prey body condition was lower than the population average [42,75,76]. This is a possible outcome of the habitat selection shown by fallow deer in San Rossore Estate, where fenced open pastures are the richest in trophic resources. These areas are also the riskiest in terms of exposure to predation but represent optimal feeding places. Therefore, deer using these areas accepted a high predation risk to benefit from high-quality forage; this was shown to increase the reproductive success of adult males [77,78] and more in general, the use of higher quality feeding areas can lead to more satisfactory body condition. In other environmental conditions, such as, for example, in the Białowieża Forest, prey avoided overlapping areas with the wolf [14], while in San Rossore Estate the fallow deer ran the risk of spatial overlap with the wolf, especially in the areas important from a trophic point of view, changing their anti-predator behaviour in accordance with the habitat and the season [79].

## 5. Conclusions

The high behavioural plasticity of wolves has led these predators to occupy noisy and human-modified environments such as areas close to large cities or including a number of human settlements. In the presence of prey populations, wolves can expand their hunting territory to these peri-urban areas.

In a Mediterranean habitat, where very dense scrubland can be found and ungulates can find shelters, a lone wolf specialised in killing prey in open areas using landscape elements (i.e., fences) to increase the success rate of his attacks. We cannot consider our findings to be universal and applicable to every case since this research only looks at the behaviour of a lone wolf. This form of behaviour has undoubtedly been promoted by the environment and the availability of prey, underscoring once more the tremendous behavioural plasticity of this species. We found kill sites in the richest areas in terms of food resources for prey [71], and it seems that fallow deer were willing to accept the highest predatory risk in order to use these areas to reach better body conditions. After all, the areas richest in food supply were those in which the likelihood of encountering a greater density of prey was higher [71]. In this respect, our study revealed that the kill sites were in the locations with the highest density of prey [14]. Thus, wolf predation on the fallow deer community was not targeted at young (less than 1-year-old), debilitated, or aged individuals, but mainly at fit individuals (males and females) aged between 1 and 3 years.

## Figures and Tables

**Figure 1 animals-13-00480-f001:**
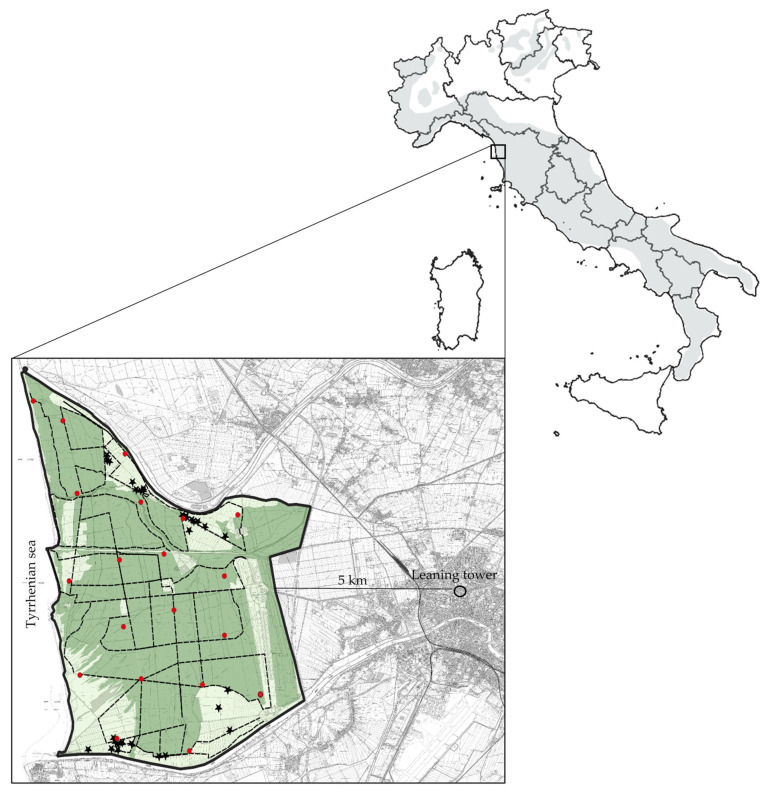
Study area. The large box shows the boundaries of the protected area (bold line) and the distance from the world-renowned leaning tower in Pisa, Italy. Dark green represents wooded areas, while light green represents open areas, i.e., meadows and marshes. Dashed lines represent transects used for searching predations, red dots represent camera trap locations, and black stars represent kill sites. The large map shows the wolf distribution in the Italian peninsula (in grey) according to ISPRA [46] and the position of the study area (small square).

**Figure 2 animals-13-00480-f002:**
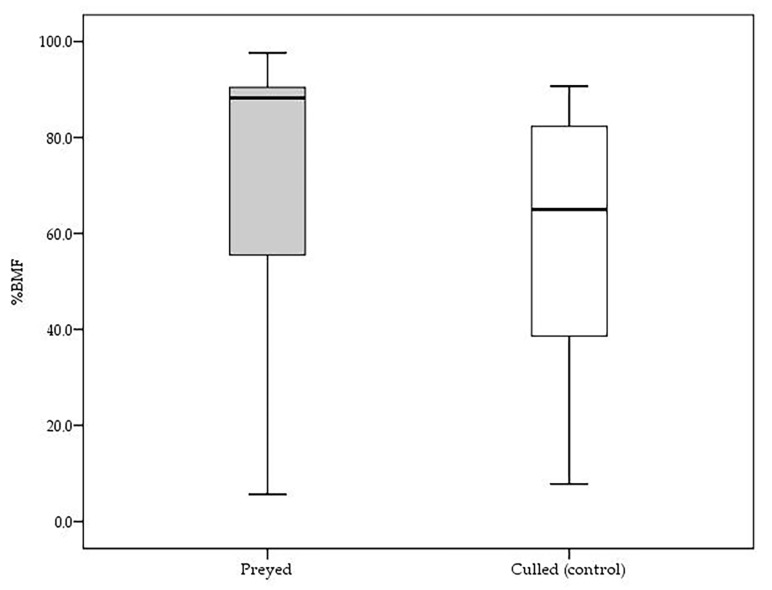
Metatarsus bone marrow fat percentage (%BMF) of fallow deer killed by the wolf (grey box) and culled by gamekeepers (white box).

**Table 1 animals-13-00480-t001:** Average surfaces of habitat types calculated for the kill and random sites, and relevant 0.95 confidence intervals (Lower CI and Upper CI).

Habitat Type	Kill Sites			Random Sites		
	Mean (m^2^)	Lower CI	Upper CI	Mean (m^2^)	Lower CI	Upper CI
Meadows and pastures	7006.43	6564.99	7447.86	1007.48	850.23	1164.73
Woods	571.83	231.69	911.97	5690.64	5250.61	6130.66
Marshes	35.88	10.58	61.18	354.62	306.48	402.74
Degraded coastal areas	148.83	75.88	221.78	641.57	613.89	669.25
Roads and river	87.03	70.61	103.46	155.69	49.16	262.24

**Table 2 animals-13-00480-t002:** Best models, ranked by AICc, and coefficients of resource selection function (RSF) for wolf kill sites in San Rossore Estate, Italy. Only models with a ΔAIC ≤ 2.00 are shown.

Model	Human Settlements	Coniferous Wood	Deciduous Wood	Degraded Areas	Marshes	River	Roads	Minimum Distance from Wood	Minimum Distance from Fences	Minimum Distance from Roads	df	AICc	ΔAIC
1	−0.033	−0.035	−0.025	-	-	-	−0.131	−0.002	−0.003	−0.003	8	150.7	0.00
2	−0.034	−0.035	−0.036	−0.031	-	-	−0.136	−0.002	−0.003	−0.003	9	151.7	0.97
3	−0.032	−0.034	−0.024	-	0.011	-	−0.138	−0.002	−0.003	−0.003	9	152.0	1.26
4	−0.033	−0.036	−0.026	-	-	0.005	−0.131	−0.002	−0.003	−0.003	9	152.7	2.00

**Table 3 animals-13-00480-t003:** Minimum distances of predation sites (n = 84) and random sites (n = 1000) from landscape elements present in the study area, and relevant 0.95 confidence intervals (Lower CI and Upper CI).

Distance from (m)	Predation Sites			Random Sites		
	Mean	Lower CI	Upper CI	Mean	Lower CI	Upper CI
Fence	90.1	40.8	139.4	2563.6	2054.2	3082.1
Building	897.5	824.4	970.5	2014.6	1642.1	2387.2
River	376.1	307.4	444.8	2784.9	2477.6	3092.3
Road	211.1	155.8	266.4	1126.4	1017.8	1235.0
Wood border	157.2	118.9	195.5	62.4	46.6	78.2

**Table 4 animals-13-00480-t004:** Minimum distances from landscape elements recorded in kill sites in different seasons. In brackets, relevant 0.95 confidence intervals (Lower CI–Upper CI) are shown.

Season	Mean of Minimum Distance (m)				
	Building	Fences	River	Road	Wood Bounder
Spring	741.1 (414.2–1068.0)	53.0 (4.7–101.2)	292.7 (171.2–414.2)	233.3 (−29.1–495.8)	296.1 (225.7–449.6)
Summer	1000.5 (900.4–1109.5)	38.7 (14.1–63.4)	386.8 (316.1–457.5)	212.1 (111.9–312.2)	142.6 (98.4–186.8)
Autumn	871.3 (746.7–995.9)	121.4 (70.8–172.0)	366.6 (242.8–490.4)	189.9 (115.8–263.9)	137.0 (111.4–162.6)
Winter	861.1 (685.2–1037.0)	122.9 (69.6–176.2)	434.0 (175.9–692.0)	256.8 (53.8–459.7)	157.3 (125.8–188.8)

**Table 5 animals-13-00480-t005:** Coefficients of logistic regression model describing the relative selection of kill sites by the wolf.

Variables	β	SE	Wald	*p*	Exp (β)
Fencing	0.005	0.001	27.664	<0.001	0.995
Building	1.038	0.227	20.954	0.310	2.824
River	1.679	0.303	30.637	0.595	5.362
Road	−0.003	0.001	16.926	<0.001	0.997
Wood border	0.737	0.448	2.704	0.100	2.089
Constant	−0.818	0.132	38.490	<0.001	0.441
Hosmer–Lemeshow test	χ^2^ = 9.507		df = 8	0.301	

**Table 6 animals-13-00480-t006:** Metatarsus bone marrow fat percentage (%BMF) in different age classes in predated and culled fallow deer. Fallow deer age classes are: 0 = fawn (less than 1 year); 1 = young (1–2 years); 2 = sub-adult (2–4. years.); 3 = adult (more than 4 years).

Age Class	n	Preyed			n	Culled			Student’s *t*-Test	
		%BMF	Lower CI	Upper CI		%BMF	Lower CI	Upper CI	t	*p*
0	7	56.0	36.3	75.7	12	29.8	17.5	42.1	1.300	0.272
1	20	83.5	77.2	89.8	28	64.0	53.8	74.3	3.335	**0.002**
2	15	71.7	56.5	86.8	29	65.7	56.8	74.7	0.759	0.452
3	6	50.1	21.7	78.5	7	52.5	32.9	72.1	−0.114	0.913

## Data Availability

Not applicable.

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
