# Peer review of "A Predator on the Doorstep: Kill Site Selection by a Lone Wolf in a Peri-Urban Park in a Mediterranean Area"

_animals, 2023, doi:10.3390/ani13030480_

Round 1

Reviewer 1 Report

The presented study is interesting and generally well written. Although it is a case study based solely on one wolf individual, it brings new data on hunting behaviour of solitairy wolves. Feeding ecology of wolves during solitairy phase of their life-histories (dispersal, "floating" etc.) is much less described in the scientific literature compared to relatively well studied hunting strategies of wolf packs. Thus, the study is valuable and may be interesting for readers. However, I have several comments and suggestions that may improve the manucript:

1) it may be unclear for readers unfamiliar with Italian wolf population how the study area is situated in relation to known wolf range. It may be a good idea to add more information on this topic in intoduction and/or study area descrtiption, e.g. how far the study area is from permanent wolf range? is it connected by ecological corridors? 

I would also suggest to add layer with the current range of the Italian wolf population to Fig.1

2) In the methods section, genetic analyses are only briefly mentioned. While there is no point in describing them in detail (referring to the Canu et al. paper is sufficient), I would suggest to add information on the type and number of used genetic markers (microsatellites? how many loci?)

Concerning genetics, I wonder if having the wolf's genotype, it is possible to identify its origin (source population or even natal pack)? Is your set of genetic markers the same as used by the Italian Institute for Environmental Protection and Research (ISPRA)? If yes, it should be possible to quickly compare the animal's genotype with their database. If the wolf's natal pack could be identified, it may partially explain the indiviual's prey selection patterns (wolves can "inherit" hunting behaviour by learning from their parents and other pack members). 

3) As the performed analyses concern a single individual, I would suggest to underline in the discussion that you have to avoid general conclusions due to great behavioural diversity observed in wolves

4) I suggest to modify the paragraph suggesting that using fences is a adaptation to hunt alone - this behaviour has been already described in pack-living wolves, see Bojarska et al. 2017 (https://www.sciencedirect.com/science/article/abs/pii/S0378112716309434)

5) A wolf living alone for such an extended period is a rather unusual case. Could you speculate why it remained in the area despite the lack of other wolves/potential breeding partners?

Similar case has been described in western Canada: https://esajournals.onlinelibrary.wiley.com/doi/abs/10.1002/ecy.2513

6) Other minor points:

- figure 2 and 3 - in the pdf file figure captions overlay with the figures and are partially illegible

- line 95-7 - provided literature citations are publications showing that ungulates are the main wolf prey in Europe, but there is no literature reference for the first part of the sentence ("wolves moved to peri-urban areas"). I suggest Kuijper et al. 2016 (https://royalsocietypublishing.org/doi/full/10.1098/rspb.2016.1625) and references within

- line 93 - another relevant pubication on wolf attacks on pets: https://www.mdpi.com/2076-2615/11/9/2497

- line 88 - citation needed for the statement "In Europe, in last decades, the wolf has been increasing" - e.g. https://www.science.org/doi/10.1126/science.1257553

- lines 475-478 - the sentence is unclear, what do you mean by the expression "even because"?

- a new paper on antipredator behaviour in fallow deer has been published recently - worth including in the discussion: https://doi.org/10.1093/cz/zoac083

Author Response

Regards

Paolo

Reviewer 2 Report

This is an interesting study as there is not much information on lone wolves. I have two major concerns, the first regarding the data quality as you did not use methods allowing you to systematically follow the wolf (telemetry or snowtracking). It is unclear how you found kills. How much on transects, how much did you see by chance from roads? If the kill sites are not completely randomly found, then you will find them nearer to roads (or whatever you used for walking) than random points. Fences are also usually built nearer to roads, so also this can be biased. You need to provide evidence that your data is in fact representative for the locations of kill sites before comparing them to random sites. My second concern is with the presentation. There is too much repetition of results in text, tables and figures, just present it once. There is also a lot of unnecessary test result reporting, the most important is to provide data, so parameter estimates along with some sort of comparison (e.g. confidence intervals or if not possible significance tests). You use too many single or two-sentence paragraphs, combine them by theme. Published information is reported in the present tense (so e.g. in line 59 not “was well…” but “is well…”). In my opinion, the information you gathered is important to provide, so I recommend to present as much as possible with data and as little as possible superfluous text. I therefore recommend to shorten the paper, improve the writing (please avoid the passive voice wherever possible) and refrain from overinterpretation of results.

Detailed comments:

Line 21. “Notorious” is usually used for negative behaviour, here probably better use “known”

Line 22: “very poor knowledge is available” is awkward, better write e.g. “little is known”

Line 31-32. The data do not allow concluding an “adoption of a high risk feeding strategy by deer”.

Line 37 (and 296). Incorrect use of n, 82 is the number of fallow deer you found, not the sample size. N would be the number of all prey if you calculated a proportion. You can only conclude there is no attack on domestic ungulates if you followed the wolf (e.g. by snow tracking), the wolf might have tried to attack domestic ungulates, but unsuccessfully.

Line 41. Not “run” but “ran”

Line 44. How do you know that “the wolf pushed fallow deer towards a fence to constrain them and preventing them from escaping”, videomonitoring? You missed a publication that specifically treated this behaviour (Bojarska K., Kwiatkowska M., Skórka P., Gula R., Theuerkauf J., Okarma H. 2017. Anthropogenic environmental traps: where do wolves kill their prey in a commercial forest? Forest Ecology and Management 397: 117-125).

Line 50-52. You cannot conclude with one example that this is an adaptation to human-dominated landscape. I observed a single wolf (which usually hunted in a pack) killing a red deer kill in a large forest.

Lines 62-64. Hunting in high density prey sites does not mean that they are killed there, in fact the kill sites are in areas of average prey density, so wolves might select hunting areas but not kill sites (line 83 needs also modification), they just kill there where they can (Theuerkauf J., Rouys S. 2008. Habitat selection by ungulates in relation to predation risk by wolves and humans in the Białowieża Forest, Poland. Forest Ecology and Management 256: 1325-1332).

Line 110. The SI unit is km², not hectares. What kind of protected area, a hunting reserve?

Line 133. “no other wolf” instead of “no more wolves”.

Figure 1. The figure does not provide any important information. It is enough to provide the distance to Pisa in the text.

Table 4. This makes only sense if changed to average and combined with confidence intervals to see if there are seasonal differences. Change m to km with one decimal. For now it does not look very informative.

Table 5. Results depend on how you found the kill sites. You should present the results either just with the regression or with AIC, not necessary to do it twice.

Line 196. No sense to provide the length to the meter, 104.9 and 8.7 km is sufficient (especially as your resolution is 50 m). It is important to provide information how you selected transects. Were these along tracks or completely random by e.g. compass?

Line 219. Why 188 random sites?

Line 223. Do you mean by “minimum mapping unit” resolution? In this case use the latter term.

Line 292. First paragraph is not necessary as the whole paper is built around this, provide the info in the methods.

Line 297. To put the lack of domestic animals into context you need to provide in the methods their density in the area.

Line 300. No need to provide the surface, everyone knows how to calculate it with the radius. However, you should not provide then the surface for the habitats but proportions as these are directly comparable and more informative. SD is no interval, replace it with CI. The whole paragraph (lines 299-306) is unnecessary, put the info in a table, this makes comparison easier. I wonder if a category “open area” would be more useful in this context, you then just have two habitats to compare.

Lines 307-324. Lengthy text that could be shortened to the essentials.

Table 1 and 2. little informative, could be deleted and the essentials provided in text.

Table 3. informative, but replace SD (does not provide important info) with CI. Decide whether to use mean or median, no sense to provide both. If median, replace the table with box-and-whisker plots. I would recommend mean with CI, so you can see if any of these variables are different between kill and random sites.

Line 364-369. Do not repeat results presented in a table in the text, just state the important findings.

Table 6. present either a table or a figure. To me it does not look like the wolf killed prey in better shape than hunters, the sample size is just too low. The age classes are not clear, need to be defined here, but to me there are no clear differences. Change SD to CI.

Line 409-415. Too much repletion of results in figures.

Fig 3. Seems like a repletion of table 6, decide either figure or table. Not clear what you are presenting here, averages? Then add CI. What is the value in presenting this info without comparing it to hunted deer?

Line 453-455. To me it seems like the proportion of deer in kills is just a little higher than the proportion of deer among the available prey. You need to compare the proportion of kills with the proportion of the species in the study area before being able to assess prey selection (also for chapter 4.2 prey selection) . It is difficult to say if the wolf has developed a strategy or if he just has more success near fences (not necessarily intentional). How would a lone wolf also be able to drive a deer intentionally into a fence?

Line 456-458. Sentence makes no sense, please reword. In any case, the data are insufficient to estimate consumption per day as you did not follow the wolf constantly (e.g. by radiotracking or snowtracking).

Author Response

Reagrds

Paolo

Round 2

Reviewer 2 Report

The revision has clarified the methods a little, but there seems to be need of improvement as it is still not possible to assess the quality of the methods and data. I now assume that you found all kill sites on transects. If not, then provide information how many found on transects and how many by chance. If you searched kill sites on transects, then the correct way to create random points is to create them along the transects to make them representative for the areas searched, not within stratified polygons as these are not representative for the transects. If your transects are strip transects of a certain width (e.g. the maximum distance you saw a kill site from the transect line), you can use these transect polygons to create random points, if not, then create random points on the transect lines. If you keep Fig 1, then add all transects, so that the reader can see the distribution. Please also add the individual kill sites to Fig 1. I recommend to increase the number of random points, at least to 1000, then the confidence intervals also become smaller.

The presentation has not sufficiently improved. You still use too many single or two-sentence paragraphs, combine them by theme. The text is also still full of passive voice. I also still saw “ha” as area unit instead of “km²”. I strongly recommend to work on the presentation. You now provide numbers of domestic animals, but you need to present them the same way as other prey (best in numbers/km²).

Regarding the conclusions it is important to understand that kill sites are not high predation-risk areas, the high-risk areas are areas where the hunt starts! And these sites are not necessarily in the same area (see Theuerkauf & Rouys 2008). You cannot assume that the wolf hunts in the same areas in which you find kill sites. Therefore, the data do not allow any conclusion regarding high-risk areas or risk management of prey. If the videos allow you to see that the wolf intentionally pushed prey towards fences, then describe the technique in detail and add one video as proof in form of a supplement. Also Fig 2 does, in my opinion, not allow to conclude that “the preyed (better use “(de)predated”) deer were in better physical condition than the control ones” (lines 362-363). Here it is also contradictory to use a t-test (for normal distributions) and a box-and-whisker plot (used when the data are not normally distributed, which seems to be the case for the prey but not for culled deer).

At last, as the whole study is based on the lone-wolf situation, you need to present some proof that it is sure there were temporally no other wolves in the area, as this is a very unusual situation for 2 years. E.g. how many times did you record a wolf and how many times of these recordings could you unmistakably identify the wolf as the one individual? If you have hundreds of videos and it is always the same wolf (how did you recognise the individual?), this would be sufficient proof for most readers. Also provide the number of scats analysed and some quality measure (e.g. how many of them had DNA of sufficient quality to allow the identification of the one individual?).

Author Response

Regards

Paolo

Round 3

Reviewer 2 Report

The inclusion of the kill sites and the transects in Fig1 cleared some of my concerns up. I assume that the unusually large differences in distances between kill sites and random sites (Table 3) are the result of two problems here. First, the kill sites are located in two distinct restricted areas at the border of the study area. Any difference revealed by comparison with random points located all over the study area would therefore be large. However, this cannot be used for biological explanations. There must be a reason why you do not find any kills in most of the study area. Finding out which factor is behind the kill site distribution is essential before even thinking about the correct methods of analysis. As for now, the analytical methods are not appropriate and the results are an artefact of the restricted distribution of kill sites. Second, as I wrote in the last review, if you find kill sites on transects, the chance to find a kill site will decrease with distance to the transect, so you cannot compare any distance with random sites that do not have the same distribution. I therefore think you need to reanalyse your data to first find out the reason for the unusual distribution of kill sites and then chose an analysis that will provide robust and unbiased results. At the moment, I do not think that the results are reliable.

Author Response

Regards,

Paolo
